# A Review of Mn-Based Catalysts for Abating NO*_x_* and CO in Low-Temperature Flue Gas: Performance and Mechanisms

**DOI:** 10.3390/molecules28196885

**Published:** 2023-09-30

**Authors:** Xiaodi Li, Shan Ren, Zhichao Chen, Mingming Wang, Lin Chen, Hongsheng Chen, Xitao Yin

**Affiliations:** 1College of Materials Science and Engineering, Chongqing University, Chongqing 400044, China; lxd199611@126.com (X.L.); 20162749@cqu.edu.cn (Z.C.); 202009131184@cqu.edu.cn (M.W.); 13388961649@163.com (L.C.); yxtaj@163.com (X.Y.); 2School of Physics and Optoelectronic Engineering, Ludong University, Yantai 264000, China

**Keywords:** Mn-based catalysts, NO*_x_* catalytic reduction, CO catalytic oxidation, reaction mechanism

## Abstract

Mn-based catalysts have attracted significant attention in the field of catalytic research, particularly in NO*_x_* catalytic reductions and CO catalytic oxidation, owing to their good catalytic activity at low temperatures. In this review, we summarize the recent progress of Mn-based catalysts for the removal of NO*_x_* and CO. The effects of crystallinity, valence states, morphology, and active component dispersion on the catalytic performance of Mn-based catalysts are thoroughly reviewed. This review delves into the reaction mechanisms of Mn-based catalysts for NO*_x_* reduction, CO oxidation, and the simultaneous removal of NO*_x_* and CO. Finally, according to the catalytic performance of Mn-based catalysts and the challenges faced, a possible perspective and direction for Mn-based catalysts for abating NO*_x_* and CO is proposed. And we expect that this review can serve as a reference for the catalytic treatment of NO*_x_* and CO in future studies and applications.

## 1. Introduction

As the primary pollutants emitted from sintering, the coking and cement industries, etc., nitrogen oxides (NO*_x_*) and carbon monoxide (CO) are harmful to both the environment and human health, often leading to acid rain, photochemical smog, ozone depletion, global warming, and so on [1,2]. Consequently, many scholars have focused their efforts on exploring effective strategies for removing these pollutants [3,4,5,6]. Among the available technologies, selective catalytic reduction (SCR) using NH_3_ as the reducing agent was identified as an effective and reliable method for mitigating NO*_x_* emissions [7]. Additionally, CO catalytic oxidation is also considered as a pivotal method for reducing hazardous exhaust emissions [8]. However, the lack of good-performance catalysts restricts the extensive utilizations of NH_3_-SCR and CO catalytic oxidation technologies.

For the NH_3_-SCR reaction, V_2_O_5_-WO_3_(MoO_3_)/TiO_2_ catalysts have been commercially employed in power plants due to their excellent de-NO*_x_* performance within the temperature range of 300–400 °C. Nevertheless, the narrow working temperature window and poor SCR activity at lower temperatures have constrained the application of this method in low-temperature flue gas denitrification. Additionally, Cu-CHA catalysts have been commercially used in vehicles due to their superior SCR activity and hydrothermal stability [9]. Nevertheless, Cu-CHA catalysts are sensitive to sulfur oxides, resulting in a significant decrease in SCR activity at lower temperatures [10,11]. As for CO catalytic oxidation, noble metal catalysts, including Au [12], Pt [13,14], Pd [15], etc., have been extensively employed to remove CO. Noble metal catalysts display a higher intrinsic activity for the conversion of CO and demonstrate good resistance to sulfur poisoning at lower temperatures [16]. However, noble metal catalysts also present some disadvantages. For instance, they are chemically sensitive and might degrade rapidly in the presence of impurities [17]. Furthermore, they also show poor catalytic activity at lower temperatures [18]. Comparing transition metal and noble metal catalysts, it has been found that they could control CO almost equally, provided that the transition metal catalysts are used in larger volumes [19]. Therefore, transition metal catalysts have been considered as a promising alternative [20,21]. Among the transition metal oxide catalysts, Mn-based catalysts have attracted much attention due to their good low-temperature redox property [22]. However, Mn-based catalysts are easily poisoned by SO_2_ and H_2_O at lower temperatures and suffer from a low N_2_ selectivity at higher temperatures in the NH_3_-SCR reaction, as well as a narrow operating window [23]. Therefore, great efforts have been made to improve the catalytic performance and widen the temperature window [24,25]. Although some review articles have discussed the NH_3_-SCR reaction over Mn-based catalysts, a systematic summary of Mn-based catalysts for NO*_x_* catalytic reductions and CO catalytic oxidation is still missing. Hence, this review aimed to present a comprehensive review for the progress of Mn-based catalysts for both NH_3_-SCR reactions and CO catalytic oxidation processes. In this review, the catalytic performance, reaction mechanisms, and interaction effects of the catalytic process for NO*_x_* and CO are summarized. Furthermore, the crucial factors which influence the catalytic activity of Mn-based catalysts for NO reduction and CO oxidation are systematically identified. The structure diagram of this review is elucidated in Figure 1.

## 2. Mn-Based Catalysts for Abating NO*_x_* and CO

### 2.1. NO_x_ Removal

As low-temperature SCR catalysts, manganese oxide catalysts have exhibited great potential for NO*_x_* reduction due to their diverse crystal structures and diverse metallization valences. However, pure MnO*_x_* catalysts have the disadvantages of poor anti-SO_2_ and H_2_O catalytic activities and an inferior N_2_ selectivity, which restrict their further applications. To address these shortcomings, some researchers have made efforts to enhance the SCR catalytic performance by forming other mixed oxides, such as transition or rare earth metal oxides, regulating the morphology and structure, and introducing suitable supports for the catalysts [26,27].

#### 2.1.1. Single Mn Oxide Catalysts for NO*_x_* Removal

(1)Effect of crystal phase and morphology

The SCR catalytic performance of manganese oxide catalysts is influenced by several key factors, including the oxidation state, crystallinity, specific surface area, and morphology [28,29,30,31]. And these factors exert crucial influences on the catalysts’ catalytic performance to varying degrees. Manganese oxides are composed of octahedral [MnO_6_] units, which could form diverse arrays of tunnels and layered structures through the sharing of corners or edges [32,33]. These crystalline MnO_2_ materials (α-, β-, δ-, γ-, λ-, and ε-MnO_2_) can be categorized into three primary groups based on their structures: 1D, 2D, and 3D mesh structures, respectively. The 1D tunnel structures include α-, β-, and γ-MnO_2_, each of which feature different tunnel arrangements: 1D (1 × 1) (2 × 2), (1 × 1), and (1 × 1) (1 × 2) tunnel, respectively. ε-MnO_2_, which is similar to γ-MnO_2_, exhibits a tunnel structure with highly disordered manganese lattice points and an irregular tunnel shape. δ-MnO_2_ exhibits 2D-layered structures formed by the shared side of the [MnO_6_] octahedron. On the other hand, λ-MnO_2_ shows a typical spinel structure, characterized by a 3D (1 × 1) tunnel arrangement [34,35]. Figure 2 illustrates the schematics of different MnO_2_ crystal structures, suggesting their unique arrangements and configurations [36].

It has been reported that crystal structures could significantly influence the NO*_x_* conversion efficiency of MnO_2_ catalysts [37]. Dai et al. [38] suggested that α-MnO_2_ catalysts display a higher de-NO*_x_* activity compared to δ-MnO_2_ catalysts, primarily due to them being assigned a higher abundance of surface chemisorbed oxygen species. Yang et al. [39] indicated that the NH_3_ species adsorbed at Lewis sites on β-MnO_2_ catalysts exhibited poor reactivity with O_2_, resulting in less N_2_O formation and a lower NO conversion. And the catalytic performance of γ-, α-, and δ-MnO_2_ catalysts were also investigated (Figure 3). Furthermore, a range of MnO_2_ catalysts with various morphologies, including MnO_2_ nanorods, nanospheres, nanowire, nanotubes, nanoflower, mesoporous MnO_2_ nanosheets, and 3D mesoporous MnO_2_, were synthesized for removing NO*_x_* [40,41,42,43]. Li et al. [44] synthesized MnO_2_ nanomaterials with various morphologies via the hydrothermal method, as illustrated in Figure 4. The results indicated that the NO conversion efficiency decreased in the following order: nanospheres > nanosheets > nanorods for MnO_2_, and the removal efficiency of NO for the MnO_2_ nanosphere catalyst could reach nearly 100% over the temperature range of 200 to 350 °C. Gao et al. [45] successfully synthesized mesoporous MnO_2_ catalysts by employing KIT-6, SBA-15, and MCM-41 mesoporous silica as the templates. Among them, the mesoporous MnO_2_-KIT-6 catalyst, characterized by 3D cubic channels, exhibited the highest NH_3_-SCR activity and the widest temperature window. 

(2)Effect of valence state

Each valence state of manganese is associated with a stable oxide, including MnO, Mn_2_O_3_, Mn_5_O_8_, Mn_4_O_3_, and MnO_2_. The good de-NO*_x_* activity of MnO*_x_* catalysts at lower temperatures could be attributed to the presence of the multivalent oxidation state of manganese and high mobility of lattice oxygen [46,47,48]. Additionally, Yang et al. [48] conducted a study on the pathways of N_2_O formation during an NH_3_-SCR reaction over different valence states of manganese oxide catalysts, as depicted in Figure 5. It could be observed that the MnO_2_ catalysts yielded more N_2_O at lower temperatures. In situ DRIFTS results indicated that the most of the N_2_O was generated from the SCR reaction on the MnO_2_ catalysts at lower temperatures, and the N_2_O amounts produced via the NH_3_ oxidation of the catalyst increased with the rising temperature. Wan et al. [49] emphasized the significance of Mn^4+^ species as the primary active species in the NH_3_-SCR reaction. These findings were also confirmed by the studies of Zhang et al. [50] and Pappas et al. [51].

#### 2.1.2. Composite MnO*_x_* Catalysts for NO_x_ Removal

As for the stringent emission standards for NO*_x_*, relying solely on a single MnO*_x_* catalyst in the NH_3_-SCR system is not sufficient to meet the discharge requirements. Mixing or doping MnO*_x_* with other metal oxides can enhance the catalytic activity of MnO*_x_* catalysts, owing to the synergistic effects achieved by combining Mn and other metal elements. Transition metal oxides, such as Fe_2_O_3_, CeO_2_, Co_3_O_4_, CuO, etc., have been widely employed for the modification of MnO*_x_* catalysts [52,53,54]. The incorporation of other metal oxides usually generates abundant active sites, rich oxygen vacancies, and an excellent redox capacity, all of which contribute to the enhancement of the catalytic performance [55,56,57]. For instance, Shi et al. [58] synthesized Mn-based bimetallic transition oxide catalysts and investigated the impact of various transition metals on the catalytic activity of pure MnO*_x_* catalysts. As illustrated in Figure 6, the Co-MnO*_x_* catalyst achieved about a 95% NO conversion efficiency at 100 °C. This superior catalytic activity of the Co-MnO*_x_* catalyst was likely attributed to the unique manganese-rich surface activity. Kang et al. [59] synthesized a Ni-doped MnO*_x_* catalyst using the solvent-free doping method and evaluated its SCR performance. The result suggested that Ni doping could enhance the medium-temperature activity, obtaining a remarkable 100% NO conversion rate at 100–200 °C. Jiang et al. [60] suggested that the incorporation of Zr could improve the catalytic activity and SO_2_ tolerance of Mn-based catalysts. The characterization results revealed that the introduction of Zr strengthened the interaction between Zr and the active sites, resulting in the amorphous structure of the catalysts. Moreover, in situ DRIFTS studies displayed that the addition of Zr promoted the L-H reaction pathways at lower temperatures. Long et al. [61] indicated that the co-doping of Nb and Fe optimized the low-temperature SCR activity and N_2_ selectivity of MnCeO*_x_* catalysts, as depicted in Figure 7. 

#### 2.1.3. Supported MnO*_x_* Catalysts for NO*_x_* Removal

The choice of suitable supports has a profound impact on shaping the crystalline and catalytic performance of NH_3_-SCR catalysts. An ideal support not only provides a large specific surface area for the efficient dispersion of the active components, but also creates a favorable environment for catalytic reactions to occur. To date, extensive efforts have been dedicated to exploring various support materials, including TiO_2_ [62,63], Al_2_O_3_ [64,65,66], CeO_2_ [67,68,69], SiO_2_ [70], ZrO_2_ [71,72], and active carbon (AC) [21,73], as potential supports for the immobilization of MnO*_x_* catalysts.

(1)TiO_2_ as support

TiO_2_ as a support has demonstrated an excellent resistance to SO_2_ [74]. Moreover, it can interact with MnO*_x_* catalysts to enhance the dispersion of Mn species [75,76]. Smirniotis et al. [77] investigated the impact of different TiO_2_ phases on the SCR catalytic activity of MnO_2_/TiO_2_ catalysts. The result indicated that MnO*_x_*/TiO_2_ (hombikat, anatase) displayed the highest SCR activity, attributed to the larger specific surface area and abundant acid sites of TiO_2_ (hombikat, anatase). Li et al. [74] prepared Mn-Ce/TiO_2_-NS and Mn-Ce/TiO_2_-NP catalysts, utilizing anatase TiO_2_ with exposed {001} crystal faces (TiO_2_-Ns) and anatase TiO_2_ with exposed {101} crystal faces (TiO_2_-NP) as supports, respectively. The Mn-Ce/TiO_2_-NS catalyst exhibited higher SCR activity than that of the Mn-Ce/TiO_2_-NP catalyst, even in the presence of SO_2_, as illustrated in Figure 8. This was due to the anatase TiO_2_ {001} facets potentially preferentially reacting with SO_2_, thus avoiding the inactivation of the active sites. 

(2)Al_2_O_3_ as support

A large surface area, abundant acid sites, and a superior mechanical property makes Al_2_O_3_ an outstanding support for SCR catalysts. Yao et al. [78] synthesized MnO*_x_* catalysts with different supports, and the influence of these supports on the physicochemical properties and denitration performance of the catalysts was evaluated. The results indicated that the MnO*_x_*/γ-Al_2_O_3_ catalyst exhibited a strong NO*_x_* adsorption capacity and had abundant Mn^4+^ species, resulting in a higher SCR activity in the entire reaction temperature (Figure 9). Furthermore, comparative studies on Mn-Ce oxides supported on TiO_2_ and Al_2_O_3_ for NH_3_-SCR at low temperatures were conducted by Jin et al. [79]. The results demonstrated that the Mn-Ce/Al_2_O_3_ catalyst showed a relatively higher SCR activity than the Mn-Ce/TiO_2_ catalyst at the temperature range of 80–150 °C, primarily due to the Mn-Ce/Al_2_O_3_ catalyst having more acid sites. Li et al. [80] synthesized supported catalysts of FeO*_x_* and MnO*_x_* that were co-supported on aluminum-modified CeO_2_ for a low-temperature NH_3_-SCR reduction of NO*_x_*. It was observed that the Fe-Mn/Ce_1_Al_2_ catalyst achieved over a 90% NO conversion at 75–250 °C and displayed superior SO_2_ resistance compared to the Fe-Mn/CeO_2_ catalyst. The improved catalytic performance could be ascribed to the larger surface area, and the enhanced reducibility was due to the introduction of Al_2_O_3_. 

(3)Carbon materials as support

Carbon materials, including carbon nanotubes (CNTs), activated carbon (AC), activated carbon fiber (ACF), and graphene (GR), have been identified as attractive supports for SCR catalysts [81,82]. Su et al. [83] synthesized a range of MnO*_x_* catalysts supported by CNTs and assessed their SCR catalytic performance. The results showed that the catalyst with MnO*_x_* introduced into the CNT channels demonstrated superior SCR activity compared to the MnO*_x_* on the outside surface of the CNTs. Xiao et al. [84] reported that the denitration performance of a MnO*_x_*-CeO_2_/GR catalyst was better than that of a MnO*_x_*-CeO_2_ catalyst even in the presence of SO_2_ and H_2_O, as displayed in Figure 10. The result suggested that the introduction of GR altered the composition of the Mn species, thereby exerting a notable influence on the electron mobility. Jiang et al. [85] revealed that the introduction of AC into the catalyst resulted in an enhancement in the NO conversion efficiency. Table 1 summarizes the research results of Mn-based catalysts for NO catalytic reduction in recent years.

### 2.2. CO Removal

#### 2.2.1. Single MnO*_x_* Catalysts for CO Oxidation

Manganese oxide catalysts with different crystal structures and morphologies have exhibited significant differences in their catalytic performance for CO catalytic oxidation [86]. Xu et al. [87] reported that an α-MnO_2_ nanowire catalyst exhibited higher catalytic activity than a β-MnO_2_ catalyst, which was attributed to the α-MnO_2_ catalyst possessing a remarkable oxidation ability. Frey et al. [88] prepared non-stoichiometric MnO*_x_* catalysts and studied the relationship between their micro-structural correlation and catalytic activity for CO oxidation. The results revealed that the excellent catalytic activity of the non-stoichiometric MnO*_x_* catalyst could be attributed to the presence of nanocrystals at the ending of the nanorods. Additionally, earlier studies have indicated that the catalyst’s reactivity is linked to the ability of Mn to form different oxidation states, such as the redox of Mn^2+^/Mn^3+^ or Mn^3+^/Mn^4+^, as well as the mobility of lattice oxide species [89,90].

#### 2.2.2. Composite MnO*_x_* Catalysts for CO Oxidation

Composite oxides consist of two or more active components, and the interaction between different active species can modify their dispersion state, ultimately leading to an enhanced catalytic activity and the stability of the catalysts. In comparison to a pure MnO*_x_* catalyst, composite MnO*_x_* catalysts have preferable crystal structures and redox properties, which exhibit higher catalytic activity in CO catalytic oxidation [91]. Pan et al. [92] found that CO conversion efficiency on MnO*_x_* catalysts was significantly enhanced after introducing copper oxides. And the improved CO catalytic activity of CuMnO*_x_* catalysts was related to the resonance system of Cu^2+^+Mn^3+^ ⇆ Cu^+^+Mn^4+^ and the efficient oxidation of CO onto Cu^2+^ and Mn^4+^ species. Zhang et al. [93] prepared a range of MnO*_x_*-CeO_2_ catalysts with varying Mn/Ce molar ratios and studied the catalytic activity for CO catalytic oxidation. It could be observed that a Mn_1_Ce_1_ catalyst showed a better catalytic performance and wider operating temperature window than pure MnO*_x_* and CeO_2_ catalysts.

#### 2.2.3. Supported MnO*_x_* Catalysts for CO Oxidation

Loading MnO*_x_* on the support materials, such as TiO_2_ [94,95], Al_2_O_3_ [96], and CeO_2_ [68], provided a prospective practical application of Mn-based catalysts in CO catalytic oxidation. Dong et al. [94] designed a Mn_3_O_4_/TiO_2_ catalyst grown in situ on a titanium mesh substrate for CO catalytic oxidation. As shown in Figure 11, the Mn_3_O_4_/TiO_2_ catalyst achieved nearly complete CO conversion (100%) at a relatively low temperature of 160 °C, surpassing the performance of some noble metal catalysts. Li et al. [97] prepared CuMn/Al_2_O_3_ catalysts employing ordered mesoporous Al_2_O_3_ as a support. The result suggested that the ordered mesoporous Al_2_O_3_ led to catalysts with higher specific surface areas and large pore volumes, as well as more surface activity species, thereby enhancing the CO catalytic oxidation activity of the catalyst. A comprehensive summary of the research results of Mn-based catalysts for CO catalytic oxidation are presented in Table 2.

### 2.3. Simultaneous Removal of NO_x_ and CO

Nitrogen oxides and carbon monoxide coexist in the emissions of some plants, including coal-fired power plants, the steel industry, coking plants, and the cement industry. The development of bifunctional catalysts was of great importance for effectively removing both NO*_x_* and CO simultaneously. Manganese-based catalysts exhibit a range of oxidation states and unstable oxygen species, which play a crucial role in enhancing the adsorption and activation of NO*_x_* and CO on the catalyst surface [98,99]. In our earlier study [100], we found that γ-MnO_2_ catalysts exhibit higher catalytic activity for both NO reduction and CO oxidation compared to α-, β-, and δ-MnO_2_ catalysts (see Figure 12). And the outstanding catalytic performance of the γ-MnO_2_ catalyst could be assigned to its remarkable redox property and abundant active sites, which promote the adsorption and activation of NO and CO molecules. Zheng et al. [101] prepared CuMnO*_x_* bifunctional catalysts and evaluated their catalytic performance for NO reduction and CO oxidation. As shown in Figure 13, a Cu_1_Mn_1_ catalyst exhibited excellent activity for removing NO and CO simultaneously, achieving nearly 100% NO conversion and 96% CO conversion at 125 °C, respectively. Gui et al. [102] reported a bifunctional catalyst of Mn_2_Cu_2_Al_1_O*_x_* which possessed dual active sties and was highly active for both NH_3_-SCR and CO oxidation reactions. The results indicated that CO was more easily adsorbed on the Cu active sites, while NH_3_ was more inclined to absorb on the Mn active sites, which enabled the simultaneous occurrence of NO catalytic reduction and CO oxidation on the catalyst surface. Guo et al. [53] synthesized a CuMn-HZSM-5 catalyst via the impregnation method. The optimized catalyst achieved a 90% NO removal efficiency and nearly a 100% CO conversion rate at 200 °C. The results of the bifunctional catalysts for removing both NO*_x_* and CO simultaneously are concluded in Table 3.

## 3. Mechanisms and Interactions of NO*_x_* Catalytic Reduction and CO Catalytic Oxidation 

### 3.1. Pathways and Mechanisms of NO_x_ Catalytic Reduction on Mn-Based Catalysts

Understanding the pathways and mechanisms of NO*_x_* catalytic reductions over the catalysts was of significance in attaining efficient denitrification. In the NH_3_-SCR reaction, the primary pathways of NO*_x_* reduction could be outlined using Equations (1)–(5):(1)$${4\mathrm{NO}+4\mathrm{NH}}_{3}{+O}_{2}\to {4N}_{2}{+6H}_{2}O$$
(2)$${4\mathrm{NH}}_{3}{+2\mathrm{NO}+2\mathrm{NO}}_{2}\to {4N}_{2}{+6H}_{2}O$$
(3)$${2\mathrm{NO}}_{2}{+4\mathrm{NH}}_{3}{+O}_{2}\to {3N}_{2}{+6H}_{2}O$$
(4)$${6\mathrm{NO}}_{2}{+4\mathrm{NH}}_{3}\to {7N}_{2}{+12H}_{2}O$$
(5)$${6\mathrm{NO}+4\mathrm{NH}}_{3}\to {5N}_{2}{+6H}_{2}O$$

Among them, Reaction (1) was referred to as the “standard SCR” reaction, containing a stoichiometry with identical amounts of NO and NH_3_. In the presence of NO_2_, Reaction (2) proceeded at a higher rate compared to the “standard SCR”, so it was defined as “fast SCR”. When an excess of NO_2_ (NO_2_/NO > 1) was present in the flue gas, Reactions (3) and (4) happened. Reaction (5), between NH_3_ and NO, proceeded in an oxygen-free or low-oxygen atmosphere. 

The Eley–Rideal (E-R) and Langmuir–Hinshelwood (L-H) mechanisms are commonly accepted pathways in the NH_3_-SCR reaction [105], as illustrated in Figure 14. As for the E-R mechanism, ammonia molecules are initially adsorbed at the acid sites on the catalyst surface, leading to the formation of intermediates, such as -NH_2_ species and adsorbed NH_3_ species. Subsequently, these intermediates react with gaseous NO and NO_2_, ultimately resulting in the generation of N_2_ and H_2_O. The reaction process can be described using Equations (6)–(10):(6)$${\mathrm{NH}}_{3\left(\mathrm{gas}\right)}\to {\mathrm{NH}}_{3\left(\mathrm{ads}\right)}$$
(7)$${\mathrm{NH}}_{3\left(\mathrm{ads}\right)}{+M}^{n+}=O\to {\mathrm{NH}}_{2\left(\mathrm{ads}\right)}{+M}^{\left(n-1\right)+}-O-H$$
(8)$${\mathrm{NH}}_{2\left(\mathrm{ads}\right)}{+M}^{n+}=O\to {\mathrm{NH}}_{\left(\mathrm{ads}\right)}{+M}^{\left(n-1\right)+}-O-H$$
(9)$${\mathrm{NH}}_{2\left(\mathrm{ads}\right)}{+\mathrm{NO}}_{\left(\mathrm{gas}\right)}\to N_{2}{+H}_{2}O$$
(10)$${\mathrm{NH}}_{\left(\mathrm{ads}\right)}{+\mathrm{NO}}_{\left(\mathrm{gas}\right)}\to N_{2}{O+H}^{+}$$

Marbán et al. [106] found that Mn_3_O_4_/AC catalysts primarily follow the E-R mechanism in the NH_3_-SCR reaction, in which NO_2_ and, to a lesser extent, NO react with surface-active NH_3_ species. Xu et al. [107] also proposed that the SCR reaction over a MnO*_x_* catalyst proceeds via the E-R mechanism, in which the adsorbed NH_3_ species could react with the gaseous NO. Chen et al. [52] confirmed that the E-R mechanism plays a more significant role in the SCR reaction over MnFeO*_x_* catalysts by employing the transient reaction experiments.

For the L-H mechanism, NO was adsorbed on the active sites of the catalyst to form NO*_x_* adsorbed species. Then, the adsorbed NH_3_ reacted with the adsorbed NO*_x_* species to produce N_2_ and H_2_O. The specific processes are shown in Figure 15. In general, the L-H mechanism is easier to proceed with than the E-R mechanism owing to its low activation energy [108]. Kijlstra et al. [66] proposed that the Mn^3+^ site over the MnO*_x_*/Al_2_O_3_ catalyst was the center of Lewis acid sites, and -NH_2_ species were generated via the deamination of adsorbed NH_3_ reacting with gaseous NO and adsorbed NO at the same time. That is, both the L-H and E-R mechanisms occurred. Wei et al. [109] explored the mechanism of a Mn/TiO_2_ catalyst in the NH_3_-SCR reaction via a series of experiments and DFT calculations. The result showed that the catalytic reaction pathway on the catalysts consisted of two fundamental steps, as illustrated in Figure 16. 

### 3.2. Mechanisms of CO Oxidation on Mn-Based Catalysts

The catalytic oxidation of CO is one of the most representative prototype reactions in heterogeneous catalysis, and attracts significant interest due to its extensive applications in the environmental and energy fields. At present, the proposed mechanisms for CO catalytic oxidation mainly encompass the L-H, E-R, and Mars–van Krevelen (MvK) mechanisms [110,111,112], as depicted in Figure 17. 

The L-H mechanism, as presented in Figure 17a, involves the following key steps: CO reacts with OH- on the catalyst surface, leading to the formation of formate or carbonate species. Subsequently, the adsorbed formate or carbonate species decompose to produce CO_2_ and H_2_. Then, the presence of metal catalysts facilitates the preferential adsorption of CO and promotes the easier breaking of C-H bonds in formate species. During this reaction process, the lattice oxygen does not participate in the catalytic oxidation, and the reaction occurs through the adsorption and reaction of CO and O_2_ on the catalyst surface [111,113]. Dey et al. [114] proposed that CuMnO*_x_* catalysts predominantly follow the L-H mechanism in the CO catalytic oxidation process, primarily involving the reaction of surface-activated oxygen species with adsorbed CO species to produce CO_2_. 

**Figure 17 molecules-28-06885-f017:**
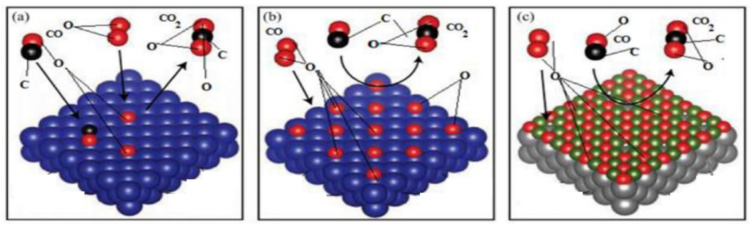
(**a**) L-H mechanism, (**b**) E-R mechanism, and (**c**) MvK mechanism diagrams for CO oxidation [115].

The E-R mechanism is displayed in Figure 17b. The mechanism involves the reaction that occurs between gaseous CO molecules and chemisorbed oxygen species (atomic oxygen and molecular oxygen). The MvK mechanism is also known as the redox mechanism, as illustrated in Figure 17c. In this mechanism, the catalyst surface exhibits a preference for the adsorption of activated CO molecules, which then react with lattice oxygen, resulting in the formation of CO_2_ and the creation of oxygen vacancies on the catalyst surface [115]. Subsequently, gaseous oxygen enters the oxygen vacancies and reacts with the partially reduced catalyst, replenishing its oxidation capacity [116]. The redox reaction mechanism involves two types of active sites: (1) active metal cation sites, which are responsible for oxidizing the reactants, and (2) active sites for the reduction of molecular oxygen. Typically, transition metal ions exhibit excellent electron conductivity, which facilitates efficient electron transfer during the redox process. Additionally, the mobility of lattice oxygen in the catalyst ensures the re-oxidation of the reduced surface, thereby enabling the regeneration of the active sites. Xu et al. [98] proposed that CO catalytic oxidation over an α-Mn_2_O_3_ nano catalyst is dominated by the L-H mechanism at lower temperatures, and turns to the MvK mechanism at higher temperatures, as shown in Figure 18. Morgan et al. [112] found a significant predominance of the MvK mechanism and a relatively minor involvement of the L-H mechanism for CO catalytic oxidation over both undoped and gold-doped CuMnO*_x_* catalysts, and the introduction of gold clearly facilitated the MvK mechanism. 

### 3.3. Interactions between Simultaneous NO_x_ Catalytic Reduction and CO Catalytic Oxidation on Mn-Based Catalysts

The simultaneous removal of NO*_x_* and CO from industrial fumes involves complex interactions between the NO*_x_* reduction and CO oxidation processes, resulting in some favorable or unfavorable consequences. Nevertheless, these interactions between multiple reactants were determined by various factors, such as the reaction temperature, the concentration of the reactants, and the catalyst properties. Understanding the interactions between various reactants and the influence of reaction conditions on the synergistic removal efficiency was of great significance in designing the catalysts for the simultaneous removal of NO*_x_* and CO.

#### 3.3.1. Effect of CO Oxidation on NO*_x_* Reduction 

Gaining insight into how CO oxidation reactants influence NO*_x_* reductions is crucial for improving the efficiency of NO*_x_* removal during the joint removal process. Nevertheless, there is currently no unanimous consensus regarding whether CO catalytic oxidation promotes or inhibits NO*_x_* reduction. Some researchers have proposed that CO catalytic oxidation promotes NO*_x_* reduction. For instance, Zeng et al. [117] confirmed that the CO oxidation reaction has a positive effect on the NO*_x_* reduction reaction. This advantageous effect could be attributed to the heat that is generated during CO catalytic oxidation, which acts as an ideal heat source to increase the flue gas temperature, thus enhancing the SCR catalytic activity at lower temperatures. Guo et al. [53] demonstrated that the introduction of CO could improve the removal efficiency of NO by facilitating NO adsorption on pre-adsorbed sites. The adsorbed CO serves as a reducing agent, converting NO to N_2_, thereby providing an alternative reaction pathway in the SCR process. Nevertheless, some scholars have suggested that CO catalytic oxidation has an inhibitory effect on NO*_x_* reduction. Gui et al. [102] found that the presence of CO has an adverse effect on the NH_3_-SCR catalytic activity of Mn_2_Cu_1_Al_1_O*_x_* catalysts. This was mainly due to the competitive adsorption of NH_3_ and CO on the active sites. Similarly, Liu et al. [118] observed a noteworthy reduction in NO conversion efficiency in the presence of CO. The decline was ascribed to the simultaneous adsorption of NO and CO on a Mn/Ti catalyst, resulting in a competitive adsorption between CO and NO, as depicted in Figure 19.

#### 3.3.2. Effect of NH_3_-SCR Atmosphere on CO Oxidation

For the simultaneous removal of NO*_x_* and CO in the NH_3_-SCR system, the CO conversion rate displays notable distinctions when compared to the individual CO catalytic oxidation reaction, suggesting that NO*_x_* might participate in the CO catalytic oxidation reaction. Zheng et al. [101] found that the CO conversion in a coordinated experiment over a Cu_1_Mn_2_ catalyst was higher than in a separate experiment, suggesting that NO played a facilitating role in the CO catalytic oxidation reaction. Guo et al. [53] indicated that the adsorption of NO on a CuMn-HZSM-5 catalyst surface generated NO_2_^−^ and N_2_O_2_^−^ species, which served as key intermediates for the oxidation of CO to CO_2_, as depicted in Figure 20.

## 4. Conclusions and Perspectives

This review provided an in-depth summary of the research progress of Mn-based catalysts in the elimination of NO*_x_* and CO. The catalytic performance, reaction mechanisms, and influence factors of Mn-based catalysts for eliminating NO*_x_* and CO were summarized. Pure MnO*_x_* catalysts exhibit a good catalytic activity for NO*_x_* catalytic reduction and CO oxidation, but with a narrow operating window and poor resistance to toxic substances. The modification of MnO*_x_* catalysts through the incorporation of other metal oxides has been demonstrated to enhance the catalytic activity and widen the operating window. Moreover, the introduction of supports, such as Al_2_O_3_, TiO_2_, and carbon materials, is also an effective strategy for improving the catalytic activity in NH_3_-SCR and CO catalytic oxidation reactions. Despite significant advancements in Mn-based catalysts for the removal of NO*_x_* and CO, there remains a pressing need for further in-depth research to develop catalysts with a higher catalytic activity for NO*_x_* reduction and CO oxidation in industrial flue gas conditions. The following aspects could be considered in the future:(1)Mn-based catalysts exhibit a poor N_2_ selectivity in the NH_3_-SCR reaction. This is primarily ascribed to the strong oxidizing property of Mn-based catalysts, resulting in the non-selective reduction of NH_3_ on the catalyst surface, thereby producing a large amount of the by-products, N_2_O. Further research should focus on improving the N_2_ selectivity. For enhancing the SCR catalytic properties, it is imperative to inhibit the non-selective catalytic reduction of NH_3_, thus enhancing the utilization rate of NH_3_.(2)The resistance to SO_2_ and H_2_O of Mn-based catalysts is insufficient in both the NH_3_-SCR and CO catalytic oxidation reactions. In future studies, scholars should concentrate their efforts on optimizing the active components and developing new structures and morphologies to avoid catalyst deactivation. Furthermore, a crucial focus should be placed on investigating the regeneration and recycling processes of the catalysts after deactivation.(3)The interaction mechanism between these two pollutants remains a controversial topic. In further studies, it is essential to employ other methods, such as DFT calculations and reaction kinetics, to gain a better understanding of the reaction processes.

## Figures and Tables

**Figure 1 molecules-28-06885-f001:**
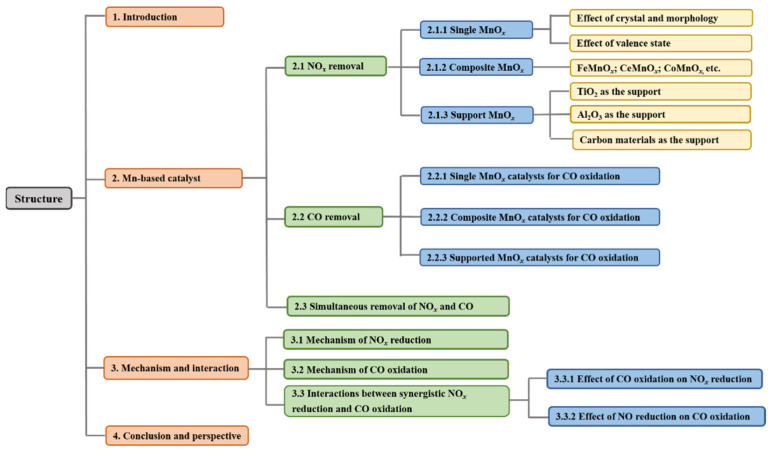
Structure diagram of this review.

**Figure 2 molecules-28-06885-f002:**
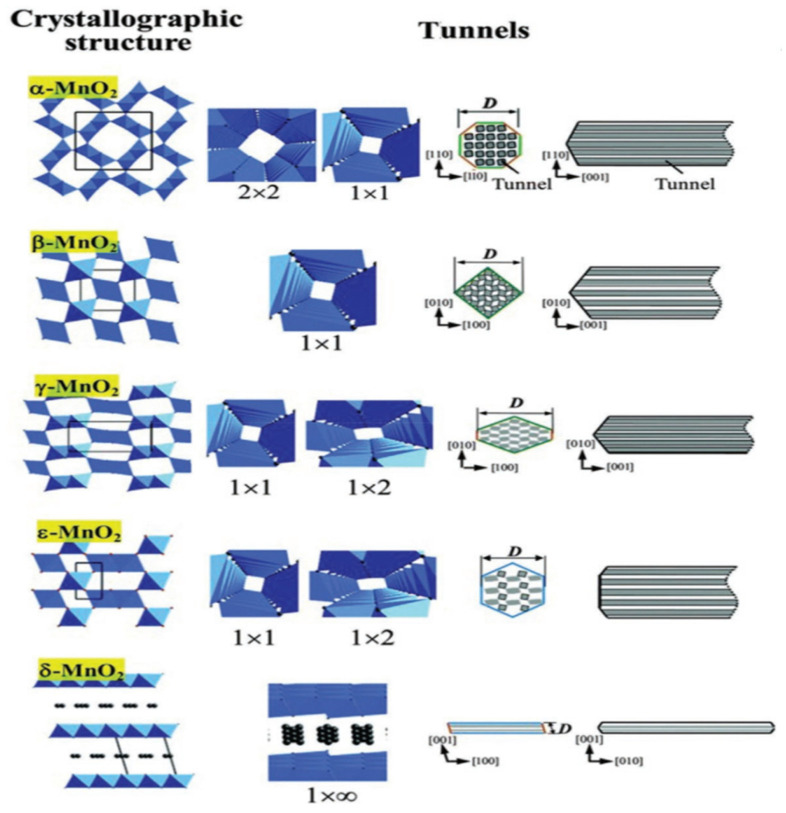
Structural models of various crystalline MnO_2_ formations [36].

**Figure 3 molecules-28-06885-f003:**
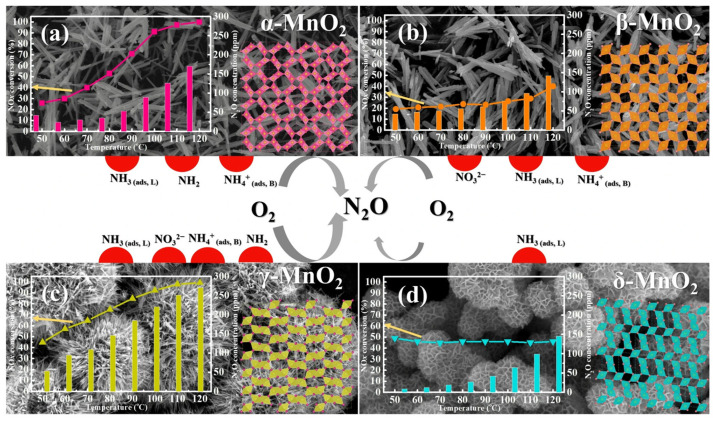
Catalytic activity and SEM diagrams of (**a**) α-MnO_2_, (**b**) β-MnO_2_, (**c**) γ-MnO_2_, and (**d**) δ-MnO_2_ catalysts [39].

**Figure 4 molecules-28-06885-f004:**
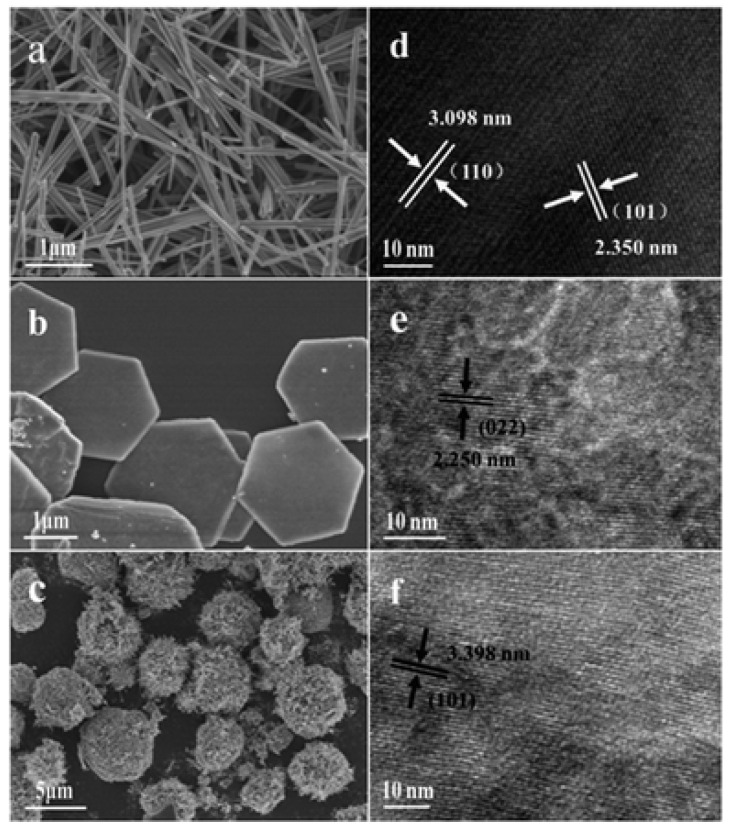
SEM and HRTEM images of (**a**,**d**) MnO_2_ nanorods, (**b**,**e**) MnO_2_ nanosheets, and (**c**,**f**) MnO_2_ nanospheres [44].

**Figure 5 molecules-28-06885-f005:**
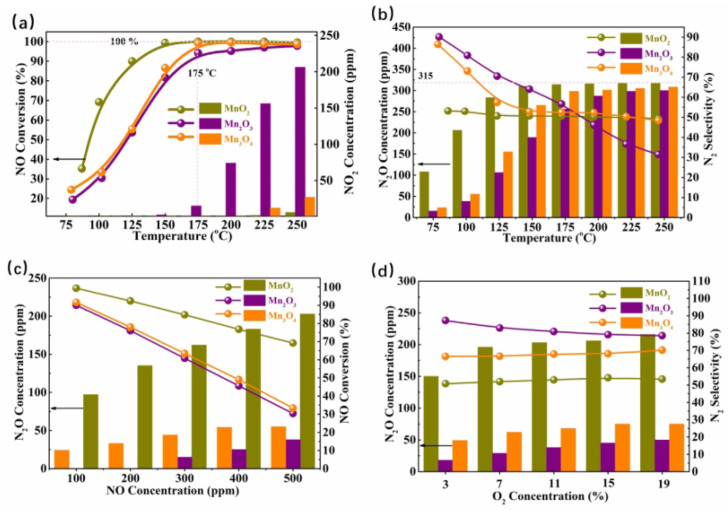
(**a**) SCR activity, (**b**) N_2_O concentration and N_2_ selectivity of catalysts at different temperature, (**c**) N_2_O concentration and NO conversion, and (**d**) N_2_O concentrations and N_2_ selectivity of catalysts at different NO and O_2_ concentrations [48].

**Figure 6 molecules-28-06885-f006:**
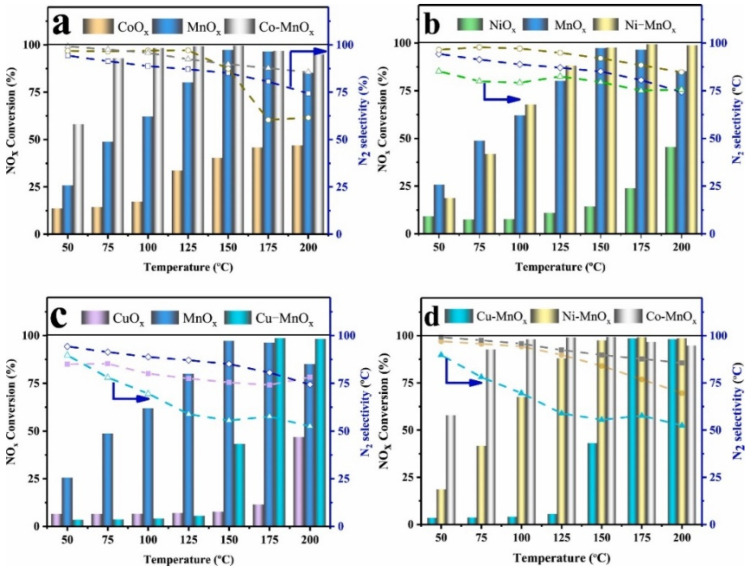
Influence of different metal oxides as additives on the catalytic activity of a MnO*_x_* catalyst: (**a**) CoO*_x_*, MnO*_x_*, and Co-MnO*_x_*; (**b**) NiO*_x_*, MnO*_x_*, and Ni-MnO*_x_*; (**c**) CuO*_x_*, MnO*_x_*, and Cu-MnO*_x_*; (**d**) Cu-MnO*_x_*, Ni-MnO*_x_*, and Co-MnO*_x_* [58].

**Figure 7 molecules-28-06885-f007:**
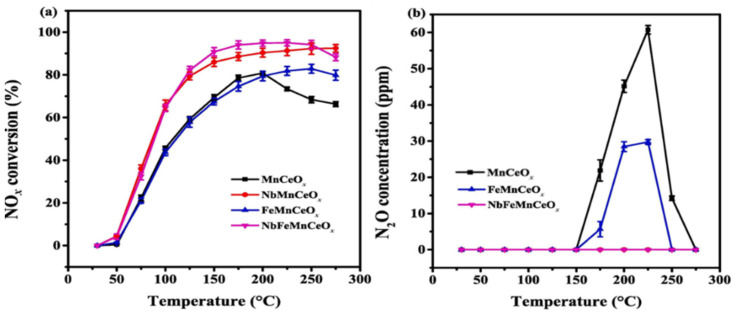
(**a**) NO*_x_* conversion and (**b**) N_2_O concentration over Mn-based oxide catalysts [61].

**Figure 8 molecules-28-06885-f008:**
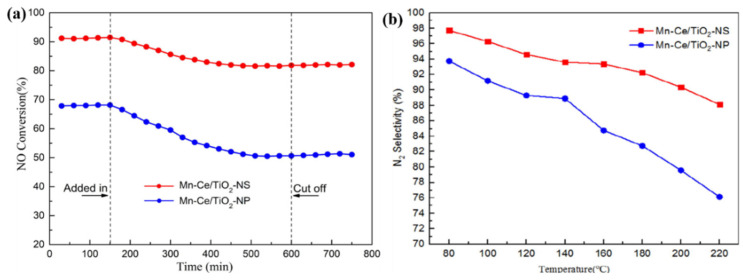
(**a**) SCR performance in the presence of SO_2_ and (**b**) N_2_ selectivity for two kinds of Mn-Ce/TiO_2_ catalysts [74].

**Figure 9 molecules-28-06885-f009:**
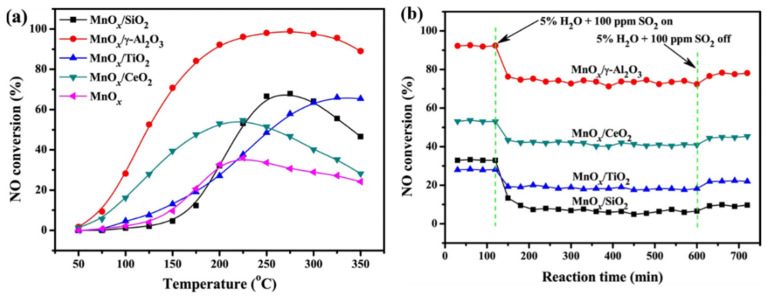
(**a**) NO conversion and (**b**) H_2_O + SO_2_ resistance at 200 °C for Mn-based catalysts with different supports [78].

**Figure 10 molecules-28-06885-f010:**
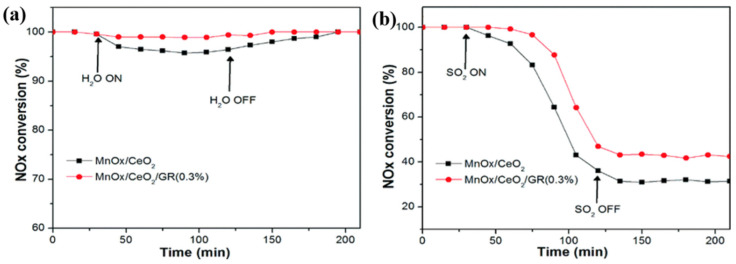
(**a**) H_2_O and (**b**) SO_2_ tolerance of MnO*_x_*-CeO_2_ and MnO_x_-CeO_2_/GR catalysts at 180 °C [84].

**Figure 11 molecules-28-06885-f011:**
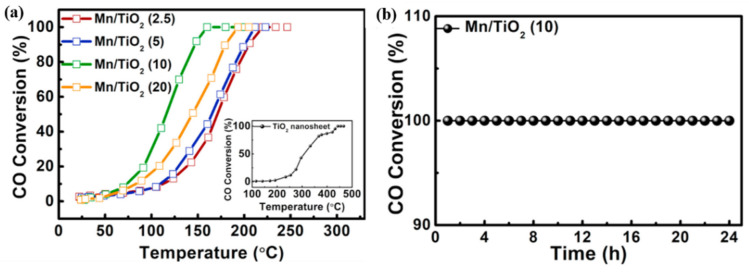
(**a**) CO conversion with temperature and (**b**) the CO oxidation stability of the Mn/TiO_2_ catalyst [95].

**Figure 12 molecules-28-06885-f012:**
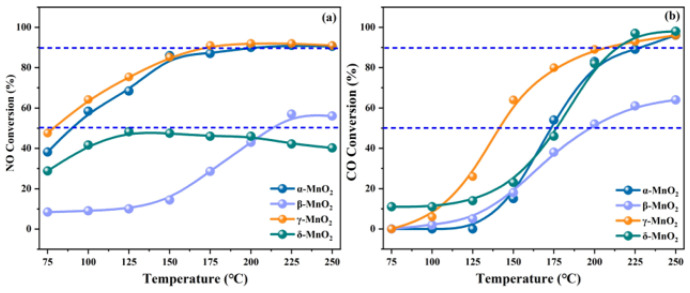
(**a**) NO conversion and (**b**) CO conversion over MnO_2_ catalysts with different crystalline phases [100].

**Figure 13 molecules-28-06885-f013:**
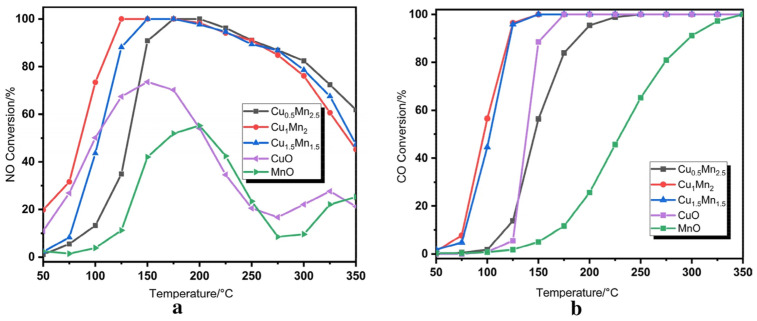
(**a**) NO conversion and (**b**) CO conversion of CuMnO*_x_* catalysts [101].

**Figure 14 molecules-28-06885-f014:**
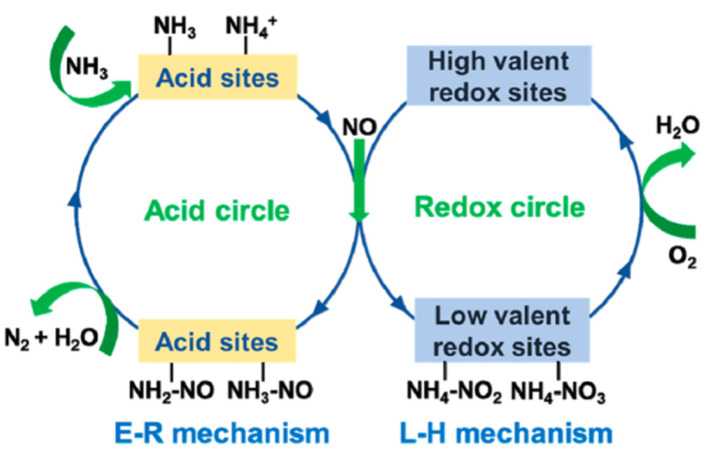
Schematic diagram of NH_3_-SCR reaction pathways over the transition metal oxide catalysts [105].

**Figure 15 molecules-28-06885-f015:**
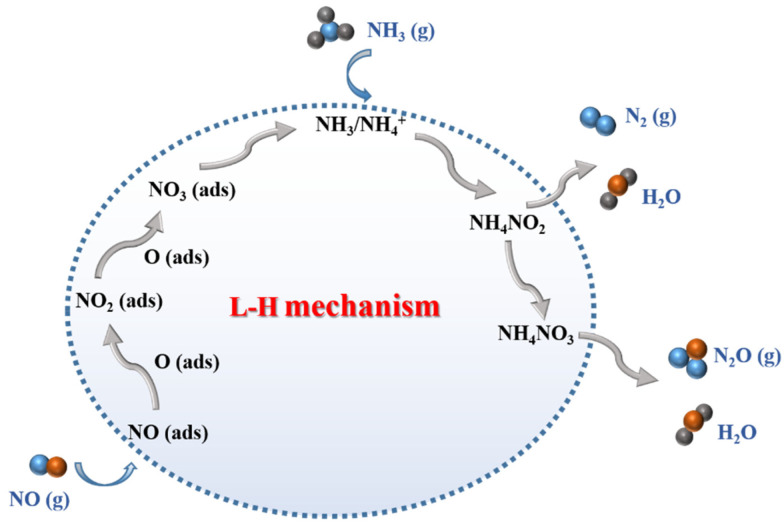
Schematic diagram of the L-H mechanism in the NH_3_-SCR reaction over the catalysts.

**Figure 16 molecules-28-06885-f016:**
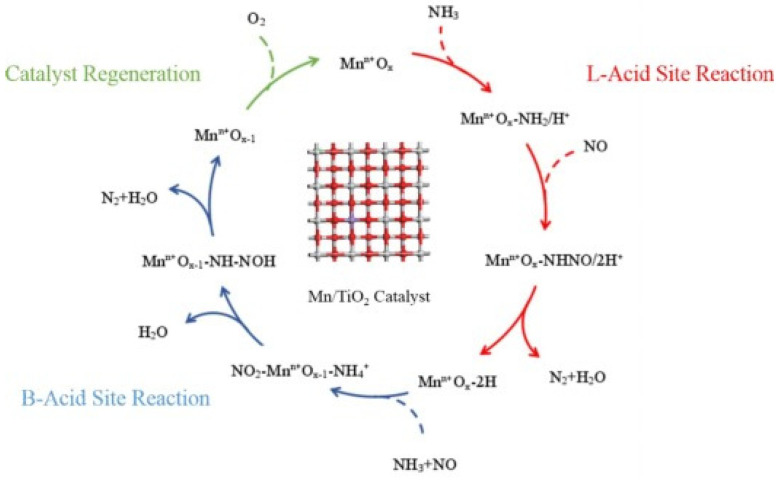
NH_3_-SCR reaction mechanism over MnO*_x_*/TiO*_x_* catalysts [109].

**Figure 18 molecules-28-06885-f018:**
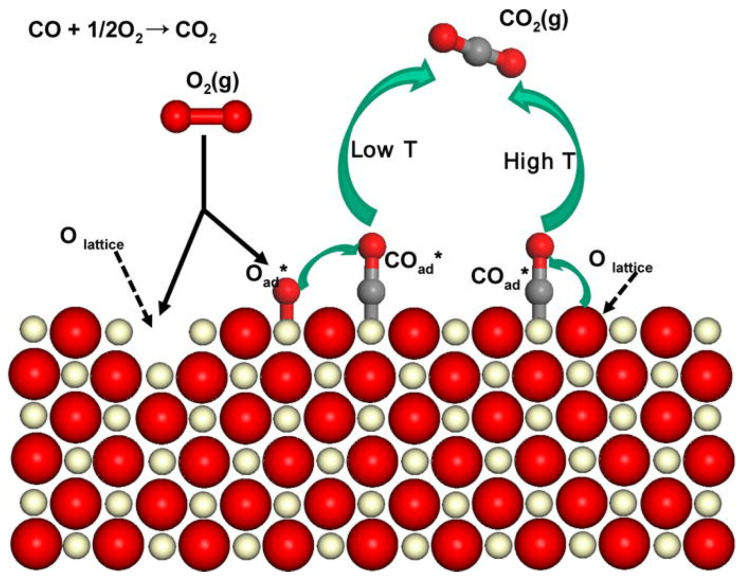
Reaction mechanism for CO catalytic oxidation on an α-Mn_2_O_3_ catalyst [98].

**Figure 19 molecules-28-06885-f019:**
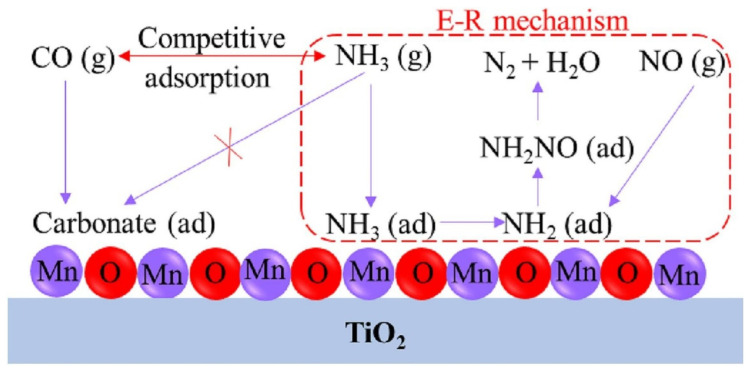
Inhibition mechanism of CO on the NH_3_-SCR reaction over a Mn/Ti catalyst [118].

**Figure 20 molecules-28-06885-f020:**
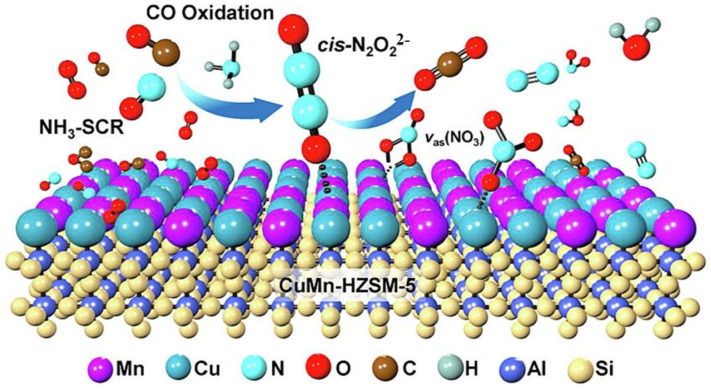
Reaction mechanisms for the selective catalytic reduction of NO and CO oxidation over a CuMn-HZSM-5 catalyst [53].

**Table 1 molecules-28-06885-t001:** Details of catalytic performance, preparation methods, and reaction conditions for Mn-based catalysts removing NO*_x_*.

Catalysts	Preparation Method	Reaction Conditions	NO*_x_*/NO Conversion (%)	T (°C)	References
α-MnO_2_	Hydrothermal	0.1% NO, 0.1% NH_3_, 2% O_2_, N_2_ as balance, 38,000 h^−1^	90%	125 °C	[38]
γ-MnO_2_	Hydrothermal	500 ppm NO, 500 ppm NH_3_, 19% O_2_, N_2_ as balance, 36,000 h^−1^	90%	100 °C	[39]
MnO_2_ nanosphere	Hydrothermal	500 ppm NO, 500 ppm NH_3_, 3% O_2_, N_2_ as balance, 28,000 h^−1^	95%	150 °C	[44]
MnO_2_-KIT-6	Impregnation	1000 ppm NO, 1000 ppm NH_3_, 5% O_2_, Ar as balance, 30,000 h^−1^	98%	100 °C	[45]
MnO_2_	Hydrothermal	500 ppm NO, 500 ppm NH_3_, 19% O_2_, N_2_ as balance, 36,000 h^−1^	100%	150 °C	[48]
Mn_3_O_4_	Hydrothermal	500 ppm NO, 500 ppm NH_3_, 19% O_2_, N_2_ as balance, 36,000 h^−1^	100%	175 °C	[48]
Mn_0.25_/TNT-H	Hydrothermal	900 ppm NO, 100 ppm NO_2_, 1000 ppm NH_3_, 10% O_2_, He as balance, 50,000 h^−1^	100%	100 °C	[51]
MnFeO*_x_*	Co-precipitation	500 ppm NO, 500 ppm NH_3_, 5% O_2_, N_2_ as balance, 75,000 h^−1^	100%	100 °C	[52]
MnCe nanowire	Hydrothermal+ co-precipitation	500 ppm NO, 500 ppm NH_3_, 5% O_2_, N_2_ as balance, 32,000 h^−1^	100%	150 °C	[44]
Co-MnO*_x_*	Solvothermal	2000 ppm NO, 2000 ppm NH_3_, 8% O_2_, N_2_ as balance, 128,000 h^−1^	100%	100 °C	[58]
NbFeMnCeO*_x_*	Co-precipitation	500 ppm NO, 500 ppm NH_3_, 11% O_2_, N_2_ as balance, 60,000 h^−1^	95%	175 °C	[61]
Mn/γ-Al_2_O_3_	Sol-gel	500 ppm NO, 500 ppm NH_3_, 5% O_2_, N_2_ as balance, 60,000 h^−1^	95%	200 °C	[78]
Mn-Ce/Al_2_O_3_	Impregnation	800 ppm NO, 800 ppm NH_3_, 3% O_2_, N_2_ as balance, 120,000 h^−1^	90%	180 °C	[79]
FeMn/CeAl	Impregnation	500 ppm NO, 500 ppm NH_3_, 5% O_2_, N_2_ as balance, 30,000 h^−1^	100%	100 °C	[80]
Ce-Mn/AC	Impregnation	500 ppm NO, 500 ppm NH_3_, 5% O_2_, N_2_ as balance, 30,000 h^−1^	95%	175 °C	[81]
Mn/CNT	Impregnation	0.08% NO, 0.08% ppm NH_3_, 5% O_2_, A_2_ as balance, 35,000 h^−1^	95%	200 °C	[83]
MnO*_x_*-CeO_2_/GR	Hydrothermal	500 ppm NO, 500 ppm NH_3_, 5% O_2_, N_2_ as balance, 24,000 h^−1^	100%	200 °C	[84]
Mn-Fe/Z-AC	Hydrothermal	450 ppm NO, 450 ppm NH_3_, 5% O_2_, N_2_ as balance, 2,000 h^−1^	98%	125 °C	[85]

**Table 2 molecules-28-06885-t002:** Details of catalytic performance, preparation methods, and reaction conditions for Mn-based catalysts removing CO.

Catalysts	Preparation Method	Reaction Condition	Best CO Conversion (%)	T (°C)	Reference
MnO*_x_*-CeO_2_	Co-precipitation	1% CO, 20% O_2_, Ar as balance, 75,000 h^−1^	100%	175 °C	[68]
α-MnO_2_	Hydrothermal	2% CO, 98% air, 12,000 h^−1^	100%	120 °C	[86]
β-MnO_2_	Hydrothermal	1% CO, 16% O_2_, N_2_ as balance, 60,000 h^−1^	90%	169 °C	[87]
Ce-MnO_2_	Hydrothermal	1% CO, 10% O_2_, N_2_ as balance, 30,000 h^−1^	100%	175 °C	[89]
Cu-MnO*_x_*	Hydrothermal	1% CO, 0.6% O_2_, He as balance, 150,000 h^−1^	100%	150 °C	[92]
Mn_3_O_4_/TiO_2_	Urea-assisted deposition	1% CO, 20% O_2_, He as balance, 7200 h^−1^	100%	150 °C	[94]
CuMnO*_x_*/γ-Al_2_O_3_	Sol-gel + co-precipitation	2.5% CO, air as balance, 30,000 h^−1^	100%	120 °C	[96]
CuMn-Al_2_O_3_	Co-precipitation	1% CO, air as balance, 10,000 h^−1^	100%	120 °C	[97]

**Table 3 molecules-28-06885-t003:** Details of catalytic performance, preparation methods, and reaction conditions for Mn-based catalysts removing NO*_x_* and CO.

Catalysts	Preparation Method	Reaction Conditions	NO*_x_* Conversion (%)	CO Conversion (%)	T (°C)	Reference
CuMn-HZSM-5	Impregnation	500 ppm NO, 500 ppm NH_3_, 5000 ppm CO, 5% O_2_, N_2_ as balance, 120,000 h^−1^	90%	100%	200 °C	[53]
γ-MnO_2_	Hydrothermal	500 ppm NO, 500 ppm NH_3_, 1000 ppm CO, 11% O_2_, N_2_ as balance, 90,000 h^−1^	91%	80%	175 °C	[100]
Cu_1_Mn_2_	Co-precipitation	500 ppm NO, 500 ppm NH_3_, 2000 ppm CO, 5% O_2_, N_2_ as balance, 100,000 h^−1^	96%	100%	125 °C	[101]
Mn_2_Cu_2_Al_1_O*_x_*	Aqueous miscible organic solvent treatment	500 ppm NO, 500 ppm NH_3_, 5000 ppm CO, 5% O_2_, Ar as balance, 80,000 h^−1^	97%	100%	200 °C	[102]
MnCuCeO*_x_*/γ-Al_2_O_3_	Impregnation	300 ppm NO, 300 ppm NH_3_, 3000 ppm CO, 16% O_2_, N_2_ as balance, 25,000 h^−1^	100%	100%	200 °C	[103]
Mn_2_Co_1_O*_x_*/IM	Hydrothermal	500 ppm NO, 500 ppm NH_3_, 5% O_2_, 5000 ppm CO, A_2_ as balance,	98%	100%	200 °C	[104]

## Data Availability

Not applicable.

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
