# Peer review of "A Review of Mn-Based Catalysts for Abating NOx and CO in Low-Temperature Flue Gas: Performance and Mechanisms"

_molecules, 2023, doi:10.3390/molecules28196885_

Round 1

Reviewer 1 Report

The review of Hongsheng Chen and co-authors is about mn-based catalysts for small molecules activation (nitrogen oxides and CO). The review consists of almost 32 pages with 123 relevant references. The review has a structure diagram which helps to navigate and understand manuscript.

The authors made a really great job. The review has critical information, advices to early career researchers, trends.

I randomly checked over 20 references - they were all relevant, leading to a reliable article, as mentioned in the text (but without doi it was quite difficult to do).

I have one remark, lines 40-41: “However, the high price and poor anti-sintering ability impeded the large-scale industrial use of noble metal catalysts.” It is not enough to explain the interest in transition metals as catalysts only by their lower initial price. The turnover of all noble catalysts does not exceed several tons. Please take a look at this recent work (you don't have to cite it! I'm not the author of this work!) https://pubs.acs.org/doi/10.1021/acs.organomet.3c00153 where the cost of catalysis is compared. Perhaps it is worth mentioning a more important reason for the interest in manganese catalysts then only the price.

Line 382, equations 11-16:  perhaps text editing changed, because it is difficult to understand. Can replace these equations with one scheme?

I certainly recommend accepting thie review for publication in the journal Molecules. 

Author Response

Comments: The review of Hongsheng Chen and co-authors is about Mn-based catalysts for small molecules activation (nitrogen oxides and CO). The review consists of almost 32 pages with 123 relevant references. The review has a structure diagram which helps to navigate and understand manuscript. The authors made a really great job. The review has critical information, advices to early career researchers, trends. I randomly checked over 20 references - they were all relevant, leading to a reliable article.

Issue 1: lines 40-41: “However, the high price and poor anti-sintering ability impeded the large-scale industrial use of noble metal catalysts.” It is not enough to explain the interest in transition metals as catalysts only by their lower initial price. The turnover of all noble catalysts does not exceed several tons. Please take a look at this recent work. https://pubs.acs.org/doi/10.1021/acs.organomet.3c00153 where the cost of catalysis is compared. Perhaps it is worth mentioning a more important reason for the interest in manganese catalysts then only the price.

Authors response: Thanks for your helpful comments. After reading the reference work you provided, we recognized that the previous statement “the high price and poor anti-sintering ability impeded the large-scale industrial use of noble metal catalysts” was really not accurate. The sophisticated organic substrates and modern reagents were as expensive as catalytic amount of noble metals during the preparation of the transition metal catalysts [1]. Therefore, the detailed reasons for the interest in MnOx catalysts were revised and added in the Introduction section in the revised manuscript.

Reference:

[1]. Komarova, A.; Perekalin, D.; Nobel metal versus abundant metal catalysts in fine organic synthesis: Cost comparison of C-H activation methods. Organometallics 2023, 42 (13) 1433-1438.

Issue 2: Line 382, equations 11-16: perhaps text editing changed, because it is difficult to understand. Can replace these equations with one scheme?

Authors response: Thanks for your good suggestion. We have replaced equations 11-16 with a schematic diagram, and the scheme was added in the revised manuscript. 

Reviewer 2 Report

            I found reviewing this paper to be very frustrating. The authors claim that Mn-based catalysts are excellent for selective NOx reduction by ammonia and CO oxidation, then propose to review the literature on these topics. The problem is that, even though NOx SCR and CO oxidation are indeed important reactions, Mn-based catalysts are not, to my knowledge, used commercially for these applications because they are not very good. Investigations into MnOx catalysts is legitimate but why do we need a review of bad catalysts? Perhaps it would be appropriate to write a review that starts out admitting these are bad catalysts and then discusses what factors affect activity; however, this paper does not do a good job of that. It just lists a bunch of papers and then shows light-off curves. The reader obtains no insights into what is going on.

            The writing is also not very good. There are many grammatical mistakes and inaccurate statements throughout. First sentence of Abstract: “its” refers back to “catalysts” and is incorrect. Since this is a review, the work may have been thoroughly reviewed but certainly not “thoroughly investigated”. Introduction: “sintering, coking, and automobiles” are different in type and should not be joined by “and”. Not everyone has “acknowledged” that SCR with ammonia is the most economical technology. If one is going to refer to commercial catalysts for SCR, how could one leave out Cu-CHA, which is the catalyst used in vehicles?

Frankly, the entire Introduction is deceptively written. Perhaps some “researchers have shifted their attention” away from precious metals for CO oxidation; however, I would argue that relatively few researchers are looking at “Cu, Fe, Mn, Gr(?), and Ni”, primarily because their oxides are really not viable alternatives. The authors write: “Among them, Mn-based catalysts were considered as the most promising candidate.” What competent scientist considers Mn-based catalysts promising candidates? This statement is just not true!

Section 2.1: The authors write: “As low-temperature SCR catalysts, manganese oxide catalysts displayed excellent catalytic activity owing to its diverse crystals and corresponding metallization valences.” (again, "its" should be "their"). If this statement were true, Mn-based catalysts would be used commercially. To my knowledge, they are not. “The excellent SCR activity of γ-MnO2 catalyst was related to its strong reducing capability, abundant acid sites and chemisorbed oxygen species.” There is no evidence that MnOx is any more acidic than CaO (a solid base that also adsorbs ammonia strongly (M. V. Juskelis, J. P. Slanga, T. G. Roberi, and A. W. Peters (J. Catal., 138 (1992) 391). Regarding the valance state of the Mn, this is going to be determined by the reaction environment. You cannot choose to use MnO or Mn2O3; the catalyst will evolve in the given environment. If one adds Ni or Co to MnOx, is the catalyst still MnOx or is MnOx the support?

It is appropriate that the section on CO oxidation is only one page long, given that Mn-based catalysts are not good catalysts for CO oxidation. But why does the Introduction imply that these materials are viable alternatives to precious metals?

This is readable but there are many mistakes.

Author Response

Comments: I found reviewing this paper to be very frustrating. The authors claim that Mn-based catalysts are excellent for selective NOx reduction by ammonia and CO oxidation, then propose to review the literature on these topics. The problem is that, even though SCR and CO oxidation are indeed important reactions, Mn-based catalysts are not, to my knowledge, used commercially for these applications because they are not very good. Investigations into MnOx catalysts is legitimate but why do we need a review of bad catalysts? Perhaps it would be appropriate to write a review that starts out admitting these are bad catalysts and then discusses what factors affect activity; however, this paper does not do a good job of that. It just lists a bunch of papers and then shows light-off curves. The reader obtains no insights into what is going on.

Authors Response: Thanks for your comments. Indeed, there are some drawbacks, such as N2 selectivity, for Mn-based catalysts, but they also have obvious advantage for low-temperature catalytic ability. So we do the work to provide reference for possible commercial applications in the future. According to your comment, we have supplemented the limitations and challenges for Mn-based catalysts in the Introduction section in the revised manuscript. Furthermore, the factors that affected the catalytic performance of Mn-based catalysts were also discussed. The modified part was highlighted with red color in the revised manuscript.

Issue 1: The writing is not very good. There are many grammatical mistakes and inaccurate statements throughout. First sentence of Abstract: “its” refers back to “catalysts” and is incorrect. Since this is a review, the work may have been thoroughly reviewed but certainly not “thoroughly investigated”. Introduction: “sintering, coking, and automobiles” are different in type and should not be joined by “and”. Not everyone has “acknowledged” that SCR with ammonia is the most economical technology. If one is going to refer to commercial catalysts for SCR, how could one leave out Cu-CHA, which is the catalyst used in vehicles?

Authors response: Thanks for your careful checking. We apologize for the inaccurate statements and grammatical errors in the manuscript. We have carefully checked the full text of our manuscript, including the contents you mentioned, and revised the corresponding mistakes. Additionally, the Cu-CHA catalyst, as one of the important commercial SCR catalysts, has also been added to the Introduction section in the revised manuscript.

Issue 2: Frankly, the entire Introduction is deceptively written. Perhaps some “researchers have shifted their attention” away from precious metals for CO oxidation; however, I would argue that relatively few researchers are looking at “Cu, Fe, Mn, Cr, and Ni”, primarily because their oxides are really not viable alternatives. The authors write: “Among them, Mn-based catalysts were considered as the most promising candidate.” What competent scientist considers Mn-based catalysts promising candidates? This statement is just not true!

Authors response: Thanks for your critical comments. Indeed, the Introduction is poor and we are sorry for the careless work. As the help of your comments, we have modified the whole Introduction carefully in the revised manuscript. The modified part was highlighted with red color in the revised manuscript.

Issue 3: Section 2.1: The authors write: “As low-temperature SCR catalysts, manganese oxide catalysts displayed excellent catalytic activity owing to its diverse crystals and corresponding metallization valences.” (again, "its" should be "their"). If this statement were true, Mn-based catalysts would be used commercially. To my knowledge, they are not. “The excellent SCR activity of γ-MnO2 catalyst was related to its strong reducing capability, abundant acid sites and chemisorbed oxygen species.” There is no evidence that MnOx is any more acidic than CaO (a solid base that also adsorbs ammonia strongly (M. V. Juskelis, J. P. Slanga, T. G. Roberi, and A. W. Peters (J. Catal., 138 (1992) 391). Regarding the valance state of the Mn, this is going to be determined by the reaction environment. You cannot choose to use MnO or Mn2O3; the catalyst will evolve in the given environment. If one adds Ni or Co to MnOx, is the catalyst still MnOx or is MnOx the support?

Authors response: Thanks for your careful checking. According to your helpful comments, we have made revision to address these issues as follows. (1) The grammatical error has been corrected by replacing “its” with “their” in the revised manuscript. And the inaccurate statement “As low-temperature SCR catalysts, manganese oxide catalysts displayed excellent catalytic activity owing to their diverse crystals and corresponding metallization valences” has also been modified in the revised manuscript. (2) We apologize for the inaccurate statement in our previous manuscript regarding to the γ-MnO2 catalyst possess more abundant acid sites. In fact, the intent was to suggest that γ-MnO2 catalyst has more acid sites compared to α-, β-, and δ-MnO2 catalysts, which has also been confirmed in our previous study [2]. (3) Regarding to the description of manganese oxides catalysts, we have corrected the inappropriate statements in the revised manuscript. Additionally, incorporating Ni or Co to MnOx, MnOx should be used as the support.

References:

[2]. Yang, J.; Ren, S.; Su, B.; Zhou, Y.; Hu, G.; Jiang, L.; Cao, J.; Liu, W.; Yao, L.; Kong, M., Insight into N2O formation over different crystal phases of MnO2 during low-temperature NH3-SCR of NO. Catal. Lett. 2021, 151, 2964-2971.

Issue 4: It is appropriate that the section on CO oxidation is only one page long, given that Mn-based catalysts are not good catalysts for CO oxidation. But why does the Introduction imply that these materials are viable alternatives to precious metals?

Authors response: Thanks for your good suggestion. According to your suggestion, we have made the necessary adjustments to the description of Mn-based catalysts for CO catalytic oxidation in the revised manuscript. And we also have revised our statements to reflect the limited suitability of Mn-based catalysts for CO oxidation in the Introduction section. The modified parts were highlighted with red color in the revised manuscript.

Reviewer 3 Report

This is a nice review paper on Mn-based oxide catalysts where the authors have analyzed literature data and have commented scientifically the interpretations of experimental data. A nice survey of the NOx reduction and COx oxidation processes, reaction mechanisms and role of solid catalysts structural aspects  is given and main options for these processes are discussed for low tempertaure flue gases treatmenst.

I have no restriction to have this review paper published in its present form without modification. Literature survey looks fine to me as main papers have been cited but obviously not all.

Author Response

Comments: This is a nice review paper on Mn-based oxide catalysts where the authors have analyzed literature data and have commented scientifically the interpretations of experimental data. A nice survey of the NOx reduction and CO oxidation processes, reaction mechanisms and role of solid catalysts structural aspects is given and main options for these processes are discussed for low temperature flue gases treatment. I have no restriction to have this review paper published in its present form without modification. Literature survey looks fine to me as main papers have been cited but obviously not all.

Authors response: Thanks for your approval for the work.

Round 2

Reviewer 2 Report

As I indicated in my previous review, I do not see the value in this paper. I do not think this paper contributes to our knowledge of NOx catalysts. If the editors choose to publish this, that's fine; but that does not make this a useful paper.

Okay